# Broccoli aptamer allows quantitative transcription regulation studies *in vitro*

**Amanda van der Sijs**[1], **Thomas Visser**[1], **Pepijn Moerman**[2], **Gert Folkers**[3], **Willem Kegel**[1] *

1 Van 't Hoff Laboratory for Physical and Colloidal Chemistry, Debye Institute for Nanomaterials Science, Utrecht University, Utrecht, The Netherlands, 2 Self-Organizing Soft Matter, Chemical Engineering and Chemistry, Eindhoven University of Technology, Eindhoven, The Netherlands, 3 Utrecht NMR Group, Bijvoet Centre fo Biomolecular Research, Utrecht University, Utrecht, The Netherlands

* w.k.kegel@uu.nl

**Data Availability Statement:** The data can be found as a supplementary file.

**Funding:** Dutch Research Council (NWO) KLEIN grant. The funders had no role in study design,

## Abstract

Quantitative transcription regulation studies *in vivo* and *in vitro* often make use of reporter proteins. Here we show that using Broccoli aptamers, quantitative study of transcription in various regulatory scenarios is possible without a translational step. To explore the method we studied several regulatory scenarios that we analyzed using thermodynamic occupancy-based models, and found excellent agreement with previous studies. In the next step we show that non-coding DNA can have a dramatic effect on the level of transcription, similar to the influence of the *lac* repressor with a strong affinity to operator sites. Finally, we point out the limitations of the method in terms of delay times coupled to the folding of the aptamer. We conclude that the Broccoli aptamer is suitable for quantitative transcription measurements.

## Introduction

Transcription regulation is an important part of cellular function and signalling, which allows cells to respond to various environmental conditions and stimuli [1]. Whether a certain gene will be expressed often depends on a delicate interplay between various transcription factors. These molecules repress or activate the transcription of a gene, often by binding to their cognate sites on the DNA surrounding the transcription start site [2]. The complexity of these regulatory systems ranges from those involving a single transcription factor to intricate interplays amongst multiple proteins and numerous regulatory sites on the DNA [3–6]. Ultimately, the overarching objective is to facilitate the recruitment of RNA polymerase (RNAP) to the promoter or hinder its access to the promoter region, where one or more regulatory signals are integrated into one regulatory output. Being able to predict the regulatory outcome for specific scenarios is important if we want to understand cellular physiology, a prerequisite to effectively design synthetic gene circuits [7–9]. However, it remains a challenge to systematically vary certain conditions in *in vivo* experiments without severe impact on the cell. *In vitro* transcription could overcome these limitations. We here present an *in vitro* method that uses a fluorescent RNA aptamer to quantitatively study transcription processes.

data collection and analysis, decision to publish, or preparation of the manuscript.

**Competing interests:** The authors have declared that no competing interests exist.

## Broccoli aptamer

The key component of this method is 2xF30-Broccoli DNA, a 105-base-pair DNA molecule that generates an RNA-aptamer upon transcription [10–12]. This aptamer can bind to the small dye DFHBI-1T and induces fluorescence, enabling real-time monitoring of transcriptional activity through a fluorescent signal [12, 13]. *In vivo*, Broccoli has been applied to visualize RNA in cells and as a gene expression reporter [14–16]. Unlike assays based on reporter proteins, the fluorescence directly indicates transcription, which can be separated from a translational step. This allows for a more convenient setup with fewer components. Recent work has shown that the Broccoli aptamer can be used to monitor *in vitro* transcription, but it hasn't been applied to quantify transcription processes, including more complicated scenarios involving regulatory components [13, 17, 18]. We use the bacteriophage T7 RNA polymerase, which is a single-subunit DNA-dependent RNA polymerase that does not require additional protein factors to facilitate transcription. It has a high specificity for its consensus T7 bacteriophage promoter and elongates about five times faster than *E. coli* RNAP. Compared to multi-subunit prokaryotic and eukaryotic RNA polymerases, this protein makes a robust workhorse for method development [19–24]. T7 RNAP is also used in recombinant protein expression systems, such as the pET expression vectors for *E. coli* [25]. Additionally, T7 RNAP is a commonly used protein in synthetic systems, such as the PURExpress system, which makes it also relevant in light of studies towards synthetic cell systems [26, 27].

## Lac repressor scenario

To explore the utility of the method for measuring the effect of transcription factors, we focus on a simple repression scenario using the *lac* repressor (LacI), one of the most widely studied transcription factors [28]. This protein is part of the *lac* operon, a set of genes in bacteria that are responsible for the metabolism of lactose and operate under a single promoter. The lactose repressor inhibits the expression of the operon by binding to an operator sequence overlapping with the promoter [29, 30]. However, by binding lactose derivatives, structural changes are induced which lead to a lower affinity for the operator [31]. In synthetic systems, the addition of isopropyl *β*-d-1-thiogalactopyranoside (IPTG), a mimic of allolactose, is used to induce transcription. Its presence prevents the repressor from binding to its specific operator site, allowing unrestricted transcription by RNAP [32]. LacI is a homotetramer, able to bind to two operators simultaneously which can lead to DNA looping as a special repression mechanism [3, 30, 33, 34]. In this paper, we use one O1 binding site placed directly downstream of the T7 promoter. Transcription is then inhibited when the *lac* repressor is bound to the operator [35, 36].

## Thermodynamic model for transcription regulation

Occupancy-based models of gene regulation have been extensively used to describe and predict outcomes of regulatory networks, with successful *in vivo* predictions of fold changes in gene expression, achieved by computing weights of configurational states of the promoter [37–43]. In these models, the main assumption is that the thermal distribution of occupied promoter sites by RNAP determines the transcription rate [5, 42, 44, 45]. Though there can be various types of interventions between RNAP binding and the complete transcription of the gene, the simplicity of the model makes it useful to interpret and predict many scenarios [40]. Recently it has been shown that an occupation-based model with simple repression can naturally lead to sustained oscillations [46]. To calculate the occupancy of the promoter, and thus the level of gene expression, we treat the DNA as a binary lattice to which proteins can bind and take the statistical weights of all the possible promoter states into account (Fig 1B). The

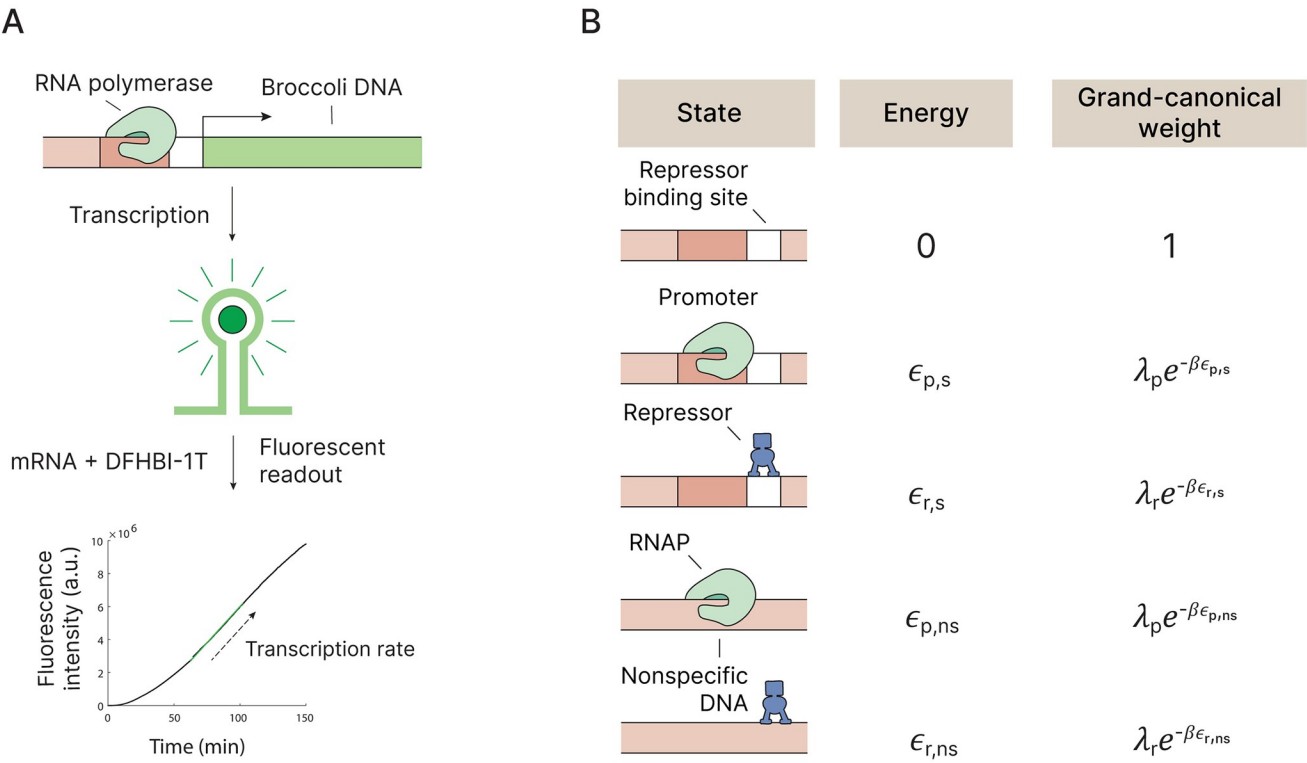

**Fig 1. In vitro transcription method and theory.** A: Schematic representation of the method. The highlighted part of the transcription curve represents the linear part of the data that is used to determine the transcription rate. B: Various regulational states and their corresponding statistical weights.

grand canonical partition function for a single gene is given by Eq (1). $\lambda_p$ and $\lambda_r$ represent the fugacities of the polymerase and the repressor, where $\lambda_i = \exp(\beta\mu_i)$ with the chemical potential $\mu_i$ of molecule $i$ and $\beta = (k_B T)^{-1}$, where $k_B$ is the Boltzmann constant and $T$ is the absolute temperature. Formally the fugacity is a Lagrange multiplier that couples to the constraint that total concentration (of RNAP, TF) is conserved. The physical interpretation of fugacity is the free, that is, unbound concentration of RNAP or TF in solution. In our framework, the fugacities are obtained by implication of mass balance in Eqs (4) and (5). $\epsilon_{s,p}$ and $\epsilon_{s,r}$ denote the specific binding free energies of the polymerase and repressor to their respective binding sites.

$$\Xi_s = 1 + \lambda_p e^{-\beta\epsilon_{s,p}} + \lambda_r e^{-\beta\epsilon_{s,r}} \tag{1}$$

The average fraction of promoter sites occupied by RNA polymerase can then be expressed as:

$$\theta_s = \frac{\lambda_p}{\Xi}\frac{\partial\Xi}{\partial\lambda_p} = \frac{\lambda_p e^{-\beta\epsilon_{p,s}}}{1 + \lambda_p e^{-\beta\epsilon_{p,s}} + \lambda_r e^{-\beta\epsilon_{r,s}}} \tag{2}$$

A more convenient quantity to measure the effect of repressors on transcription is the fold change, defined as the transcription rate in the presence of repressors divided by the reference

state without repressors:

$$\text{fold change} = \frac{\theta_s(\lambda_p, \lambda_r)}{\theta_s(\lambda_p, 0)} = \frac{1 + \lambda_p e^{-\beta\epsilon_{p,s}}}{1 + \lambda_p e^{-\beta\epsilon_{p,s}} + \lambda_r e^{-\beta\epsilon_{r,s}}} \tag{3}$$

Previous research based on *E. coli* RNAP has used the "weak promoter limit"
($\lambda_p e^{-\beta\epsilon_{p,s}} \ll 1$) to simplify these equations by only considering the properties of the repressor
[37, 39, 47, 48]. A detailed derivation of these equations can be found in Landman *et al.* [41],
which also includes analytical solutions in the weak promoter limit. As we use T7 RNAP,
which is expected to have a higher binding affinity to its promoter than its *E. coli* counterpart,
we also take the properties of the polymerase into account [49]. To solve Eqs (2) and (3), we
thus need to calculate the fugacities of both the polymerase and repressor. Since the total num-
ber of proteins remains constant in our system, we can use two mass balance equations that
consider the various reservoirs where the proteins are partitioned. The fugacities are deter-
mined self-consistently by solving

$$c_p = c_s \theta_{p,s} + c_{ns} \theta_{p,ns} + \lambda_p / \bar{v}_{aq} \tag{4}$$

$$c_r = c_s \theta_{r,s} + c_{ns} \theta_{r,ns} + \lambda_r / \bar{v}_{aq} \tag{5}$$

where $c_p$ and $c_r$ are the total (molar) concentrations of polymerase and repressor. $c_s$ and $c_{ns}$ are
the concentrations of specific and nonspecific sites. $\bar{v}_{aq}$ is the molar volume of water. The sub-
scripts $p$ and $r$ refer to the adsorption of polymerase or repressor onto a specific ($s$) or nonspe-
cific ($ns$) site. The last terms of both mass balance equations represent unbound proteins
which we consider as being "adsorbed" onto a three-dimensional lattice of $N_{aq}$ mole water
molecules as lattice sites in a volume of $N_{aq}\bar{v}_{aq}$. There, the "adsorbed" fractions are given by $\lambda_i/$
$(1 + \lambda_i) \approx \lambda_i$, with $i = p, r$ and where we assumed, as will be verified later, that $\lambda_i \ll 1$. A Boltz-
mann factor is absent here as the aqueous state is taken as the reference state with zero binding
free energy. We assume that the binding free energy of nonspecific DNA is different from that
of a specific binding site on the DNA.

Our method allows to study various regulatory architectures. To demonstrate its utility, we
will explore three different transcriptional scenarios and show how thermodynamic models
can be used to describe transcription regulation *in vitro*. Furthermore, we quantify the influ-
ence of nonspecific DNA on transcription rate as a demonstration of a significant regulatory
effect that in principle cannot be studied independently *in vivo*.

## Materials and methods

### DNA template preparation

Using pET28c-F30-Broccoli (a gift from Samie Jaffrey, Addgene plasmid #66788 [12]) as a
template, the 2xF30-Broccoli fragment was PCR amplified with a forward primer containing
an XbaI site and a reverse primer with a BamHI site. This fragment was cloned into the corre-
sponding sites of pSF-CMV-T7 (Oxford Genetics), upstream of the T7 promoter. For the O1
constructs, an insert including both the T7 promoter and O1 binding site was synthesized by
Eurofins Genomics for insertion upstream of 2xF30-Broccoli (S1 Table. DNA sequences)
using sticky ends. The oligos were annealed by mixing the complementary strands 1:1 in 10
mM Tris (pH 8), 50 mM NaCl and 1 mM EDTA, incubating the mix for 4 minutes at 95˚C
and then slowly cooling it down to room temperature. The pSF-CMV-T7 plasmid was digested
using XbaI and BglII (Thermo Scientific™) and purified (QIAquick PCR Purification Kit). Due
to the double restriction sites of BglII, the T7 and CMV promoters were removed. The

resulting backbone was mixed with the annealed oligos and ligated using T4 DNA ligase (Thermo Scientific™) for 2 hours at room temperature, following the manufacturer's protocol. The ligated vector was then transformed in *E. coli* Dh5$\alpha$ cells and the purified plasmids (QIAprep Spin Miniprep Kit) were digested and sequenced (Eurofins Genomics) to confirm the insert. The DNA templates used for transcription were amplified by PCR (Thermo Scientific™ Taq DNA polymerase) (Biometra® Tpersonal) and purified using a QIAquick PCR Purification Kit. The primers used for the amplification can be found in S1 Table. DNA sequences. Salmon sperm DNA (Invitrogen™) was used as nonspecific DNA.

## Proteins

**Lac repressor.** We thank Daniel J.Felitsky, Dr. Ruth Saecker and Dr. M.Thomas Record, Jr, of the Department of Chemistry, University of Wisconsin-Madison, Madison, for kindly providing intact *lac* repressor. The *lac* dimers were expressed and purified according to Romanuka *et al.* [50].

**T7 RNA polymerase.** The plasmid pQE30-His-T7RNAP (a gift from Sebastian Maerkl and Takuya Ueda, Addgene plasmid #124138) was used for expressing and purifying T7 RNA polymerase [51]. It was transformed into *E. coli* BL21, and expression was induced using auto-induction [52]. The His-tagged protein was purified essentially as described before [53]. In short, cells were lysed in a buffer containing 20 mM NaPO$_4$ pH 8.0, 10 mM imidazole, 300 mM NaCl, 1 mM beta-mercaptoethanol, 0.2 mM phenylmethylsulfonyl fluoride and protease inhibitor cocktail by three freeze/thaw cycles, followed by sonication. The lysate was cleared by centrifugation for 45 minutes at 45000 $g$ at 4˚C. The cleared lysate was purified using nickel-loaded Poros-MC 20 $\mu$M (Thermo Scientific™) in the above-mentioned buffer and eluted with a 300 mM imidazole block elution. Fractions containing T7 RNA polymerase were further purified by gel filtration in a buffer containing 50 mM Tris pH 8.0, 100 mM NaCl, 6 mM MgCl$_2$ and 1 mM DTT. The purified sample was stored at a concentration of approximately 100 $\mu$M in this buffer with 25% glycerol at -20˚C. The obtained protein was essentially pure, and the sample was stable for at least 1 year.

## *In vitro* transcription

Transcription reactions were carried out in 25 $\mu$L T7 transcription buffer (40 mM Tris-HCl (pH 8.0), 25 mM NaCl, 8 mM MgCl$_2$, 2 mM spermidine-(HCl)$_3$, 5 mM DTT) supplemented with 20 $\mu$M DFHBI-1T, in a 384-well plate (Greiner). The reaction was initiated by the introduction of 1 mM ATP, 1 mM CTP, 1 mM GTP, and 1 mM UTP (Thermo Scientific™). The concentration of DNA template containing the promoter, operator and Broccoli-sequence was 10 nM in all experiments. The fluorescent signal was then measured for 1–2 hours at 34˚C using a plate reader (CLARIOstar, BMG Labtech) with intervals of 10 to 30 seconds between measurements (470–15 nm excitation, 530–20 nm emission wavelength). LacI and salmon sperm DNA were diluted with the reaction buffer and added separately to the reactions prior to the addition of the polymerase.

## Data analysis

The transcription rates were obtained from the fluorescent data by applying a linear fit to the linear part of the data. Due to the folding delay of the Broccoli aptamer, the fluorescent signal starts out in a quadratic fashion before it transitions to the linear regime. (See S2 Text. Folding delays of the Broccoli-aptamer.) The transition point depends on the transcription rate, so the window that is used for the fit differs per sample. The standard deviation of different repeats is shown in the error bars.

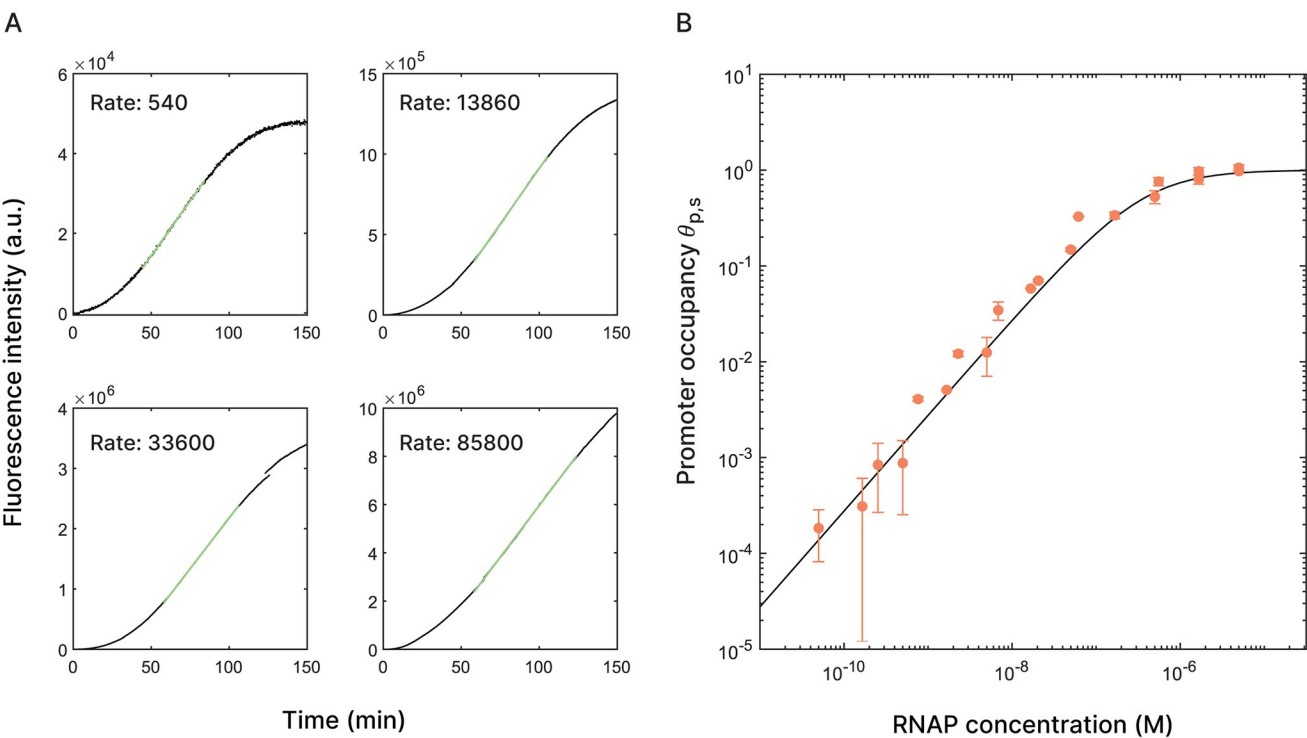

**Fig 2. Transcription rates scaled to promoter occupation.** A: Fluorescence increase for transcription experiments at RNAP concentrations of 10 nM, 100 nM, 333 nM and 3.33 $\mu$M, from low to high rates. The linear part used to determine the transcription rate is shown in green. B: The transcription rates corresponding to different concentrations of T7 RNA polymerase (pink dots), scaled to the theoretical promoter occupancy (black line). The RNAP binding free energy as found by a linear fit to the data is $\epsilon_{p,s} = -18.9 \pm 0.7\ k_B T$.

## Results

### *In vitro* transcription rates are proportional to promoter occupancy

To establish the relationship between promoter occupation by RNA polymerase and transcription rate, we studied a minimal scenario where we only varied the concentration of RNA polymerase. Some examples of the fluorescence data are shown in Fig 2A, where we highlighted the linear part of the transcription curves that are used to determine the transcription rate. The promoter occupancy by RNA polymerase follows from

$$\theta_s = \frac{\lambda_p e^{-\beta\epsilon_{p,s}}}{1 + \lambda_p e^{-\beta\epsilon_{p,s}}} \tag{6}$$

which is analogous to Eq (2), but without the presence of a repressor. As the concentration of RNAP increases, so does its fugacity and thus the fraction of occupied promoters. For these experiments, we used a DNA template consisting of the T7 promoter directly followed by the Broccoli sequence at a concentration of 10 nM. The transcription rates obtained from the experimental data were used to fit the occupancy $\theta_s$ to, with the RNAP binding free energy as the free parameter. The linear fit yielded $\epsilon_{p,s} = -18.9 \pm 0.7\ k_B T$, which is consistent with previous findings [19, 54–56]. The transcription rates and occupancy are shown in Fig 2. Depending on the experimental conditions and technique, the numbers vary from $-17.7\ k_B T$ to $-21.9\ k_B T$ [49, 54, 56, 57]. It is, however, good to note that these numbers from previous studies come from both kinetic measurements based on transcription rates, as well as equilibrium

binding experiments. In the former case, the measured binding energy will include the initiation and elongation process.

## Folding delays

As explained previously, we extract the transcription rate from the linear part of the fluorescence data. However, we observed that the fluorescent signal from the aptamer did not increase linearly during the first minutes of the reaction. This deviation from linearity can be explained by considering the time-dependent accumulation of fluorescent signal, which depends on both the rate of RNA synthesis and the rate of formation of the fluorescent RNA-dye complex. The latter consists of the folding of the RNA and the binding to the dye. As we do not know which of these processes is rate limiting, we model the conversion of inactive Broccoli RNA to the fluorescent complex as a single reaction with rate $k_f$. In S2 Text. Folding delays of the Broccoli-aptamer of the supporting information we derive an expression for the concentration of fluorescent aptamer over time

$$c_{DBa}(t) = k_p \left( t - \frac{1 - e^{-k_f t}}{kf} \right). \tag{7}$$

This equation shows that in the long time limit ($t \gg 1/k_f$), $c_{DBa} \approx k_p t$, so that the fluorescent signal is a good measure of the transcription rate. At short times ($t \ll 1/k_f$) a Taylor expansion in $k_f t$ up to the second term gives $c_{DBa} \approx \frac{1}{2} k_p k_f t^2$, indicating that the fluorescence increases initially quadratically with time, consistent with our observations. Taken together our model indicates that the shape of our fluorescent curves is consistent with the scenario where fluorescence lags behind transcription due to slow folding of the transcript. Under all conditions the transition from the quadratic to the linear regime occurs after a delay on the order of $10^3$ seconds, which implies a folding rate $k_f = 10^{-3} s^{-1}$. This time is consistent with observations by Filonov *et al.* who observed a recovery of fluorescence in 10 to 15 minutes after denaturing Broccoli with Urea and allowing it to refold in presence of DFHBI-1T [12]. In more recent work by Purhonen *et al.*, a similar delay in the fluorescent signal can be observed in their transcription experiments [18].

## Repression using LacI

As a first step towards regulation by transcription factors, we made a DNA template of 251 base pairs containing the most simple repression architecture using the *lac* repressor and a single binding site [35]. Previous theoretical studies have used *in vitro* data to show that the *lac* operon can be described using equilibrium thermodynamics [40]. For our experiments, we inserted a 21 base-pair O1 binding site between the RNAP promoter and the Broccoli sequence on our template so that transcription is prevented when the repressor is bound. Only constructs containing the O1 site showed repression, and the repression was alleviated by the presence of IPTG. (Figure in S4 Text. Template controls) Using Eqs (4) and (5), we can compute the fold change induced by the presence of repressors at various concentrations. We assume that only one protein is able to bind to either the promoter or the operator at a time, which boils down to excluded interactions within the language of the model. In previous studies, assuming the weak promoter limit, the fold change would only depend on the properties of the repressor and one could collapse all data points from various experiments onto one master curve, as shown in Landman *et al* [41]. In our case, working in the strong promoter limit, the fold change depends on both $\lambda_r$ and $\lambda_p$. In order to construct a master curve we map the

strong promoter expression onto a scaling function via

$$\text{fold change} = \frac{1 + \lambda_p e^{-\beta \epsilon_{p,s}}}{1 + \lambda_p e^{-\beta \epsilon_{p,s}} + \lambda_r e^{-\beta \epsilon_{r,s}}} = \frac{1}{1 + \lambda^*(\lambda_p, \lambda_r)} \tag{8}$$

Comparing the expressions in Eq (8) it follows that $\lambda_r^* = \frac{\lambda_r}{1 + \lambda_p e^{-\beta \epsilon_{p,s}}}$ which allows us to directly plot the statistical weight $\lambda_r^* e^{-\beta \epsilon_{r,s}}$ of all RNAP concentrations against the fold change in a master curve (Fig 3B). Using the previously determined binding energy of RNAP, the effective repressor binding energy was determined to be $\epsilon_{r,s} = -23.6 \pm 0.3\ k_B T$. This is in correspondence with values found in previous studies, which show binding energies ranging from $-23$ $k_B T$ to $-24.4\ k_B T$ for various experimental conditions [36, 41, 58–64]. In Fig 3 we also show the predicted fold change in the weak promoter limit (dotted), which lies the closest to the lowest RNAP concentration. Given the variation between the various concentrations and experimental uncertainty, the weak promoter limit is still a reasonable approximation for repression of T7 RNA polymerase.

We used *lac* tetramers for our experiments, which are in principle able to bind two operator sites simultaneously. In our current setup the DNA fragments only contain one operator and the DNA concentration was so dilute that the probability of a tetramer binding two DNA molecules was negligible. We verified this by a multi-chemical equilibrium model that includes the possibility of DNA bridging, with the result that the effect only becomes significant at DNA concentrations of two orders of magnitude beyond those in our experiments. (Figure in S3 Text. Multi-chemical equilibrium model for tetrameric repression) However, unlike the *lac* dimer, the tetramer has two binding possibilities to bind a DNA fragment. This entropic effect

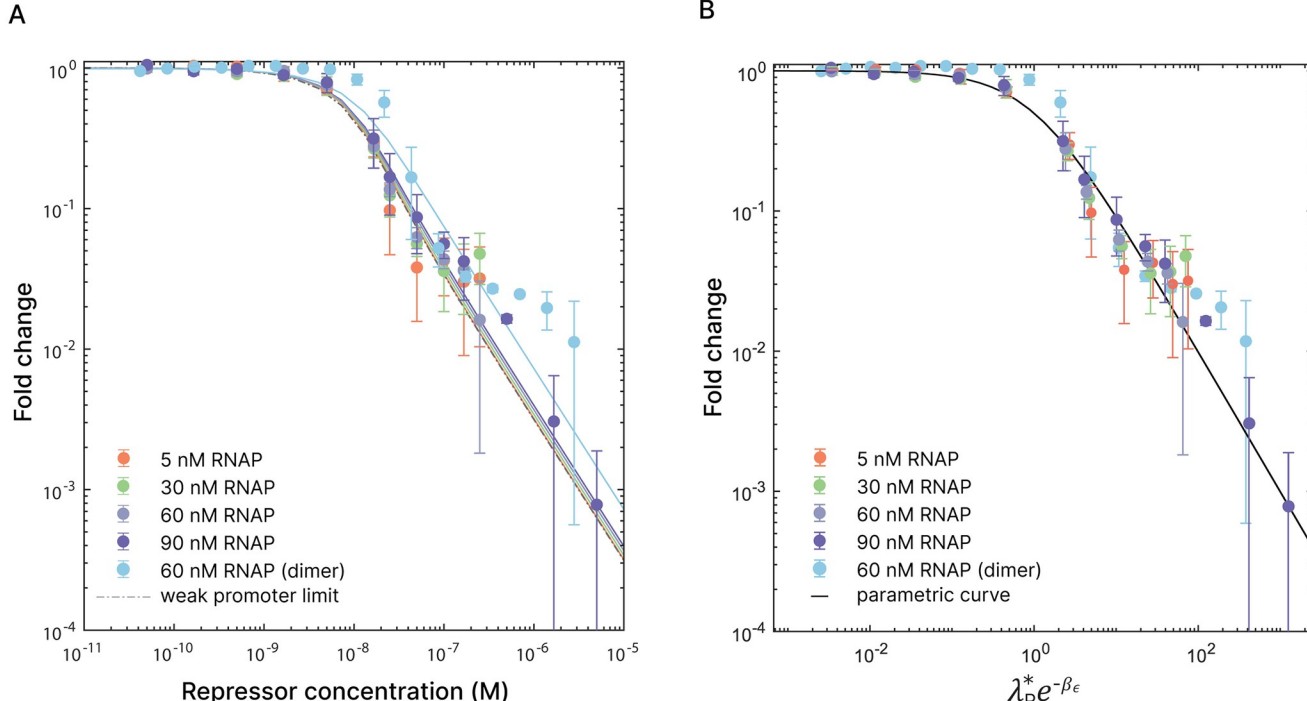

**Fig 3. Repression using the Lac operon.** A: Fold change for various repressor concentrations. The solid lines are the predicted fold changes for the different concentrations of RNAP with $\epsilon_{r,s} = -23.6 k_B T \pm 0.3\ k_B T$. The blue line shows the fold change in case of dimers, which follows from a binding energy that is $ln(2)$ lower than the energy used for tetramers. B: Data from A collapsed onto a master curve.

of $ln(2)$ is included in the effective binding energy $\epsilon_{r,s}$ that we found for the tetrameric repression. In the case of dimeric repression, this effect is absent and we can thus expect a lower effective binding energy. In Fig 3 we show the data of dimeric repression, accompanied by the expected fold change in the blue curve. Variability between protein batches might account for the differences we observed.

### Nonspecific DNA can effectively act as a repressor

It is known that both the *lac* repressor and RNA polymerase not only bind to their cognate promoters or operators, but also spend a significant amount of the time bound nonspecifically to DNA. As it is not possible to systematically study this *in vivo*, we decided to use it as a test case in our system. Previous studies have attempted to measure the binding energy of RNAP binding to nonspecific DNA, but due to very low binding energies, this was not always possible [65]. Other studies make an estimate of $< -10\ k_B T$ [66, 67]. Thus, only at sufficiently high concentrations the presence of nonspecific DNA will influence the partitioning of RNAP and hence the promoter occupancy (Eq (4)). We performed a series of experiments where we used three different RNAP concentrations to test a range of nonspecific DNA concentrations (Fig 4). These experiments were performed using salmon sperm DNA, which contains a negligible amount of promoter-like sequences specific to T7 RNAP. As the concentration of nonspecific DNA increases, the transcription rate decreases. This corresponds with the partitioning of RNAP onto the nonspecific sites, making it unavailable for specific binding at promoter sites. The number of occupied promoter sites ($c_s \theta_{p,s}$) thus decreases, following the mass balance (Eq (4)). Using the specific binding free energy of $-18.9\ k_B T$ that we determined earlier, the binding free energy that corresponds to the nonspecific DNA added to our system is $-13.4 \pm 0.6$ $k_B T$. While these numbers are specific to the DNA used in our experiments, we expect that all types of nonspecific DNA will follow the global trend in Fig 4. The theoretical fold change for these numbers is plotted for all three RNAP concentrations. If we use the fugacity multiplied by the exponent of the binding free energy as the statistical weight, we can let the data collapse onto a master curve, as shown in the inset of Fig 4. This is the first time, to our knowledge, that the effect of nonspecific DNA on transcriptional activity has been quantified.

### Discussion

In this article, we have shown that the fluorescent Broccoli aptamer can be used to quantitatively study transcription regulation *in vitro*. Due to the minimal components this approach is easily adjustable to study various transcriptional scenarios. Previously, using existing data, Landman *et al.* have reconfirmed that the fitted binding free energy of transcription factors is indeed the quantity that governs transcriptional activity, and not just an effective kinetic parameter. We applied that finding to interpret the data from our experiments and used the transcription rates to find binding free energies. In the most simple scenario, we have again confirmed that models based on the transcription rate being proportional to promoter occupancy (here by T7 RNA polymerase) are consistent with the experimental observations. We then used a minimal version of the *lac* operon to quantify repression by the *lac* repressor, paving the way to more complex regulatory architectures.

As an important illustration of what can be done using this (*in vitro*) setup that cannot be done *in vivo* without severe consequences for a cell, we showed that the presence of nonspecific DNA affects the transcription, acting effectively as a repressor in our system. Previous studies using the thermodynamic transcription regulation model have successfully taken the influence of nonspecific DNA on regulatory outcomes into account. However, the effect of changes in the availability of nonspecific DNA for transcription factors to bind to have not been included

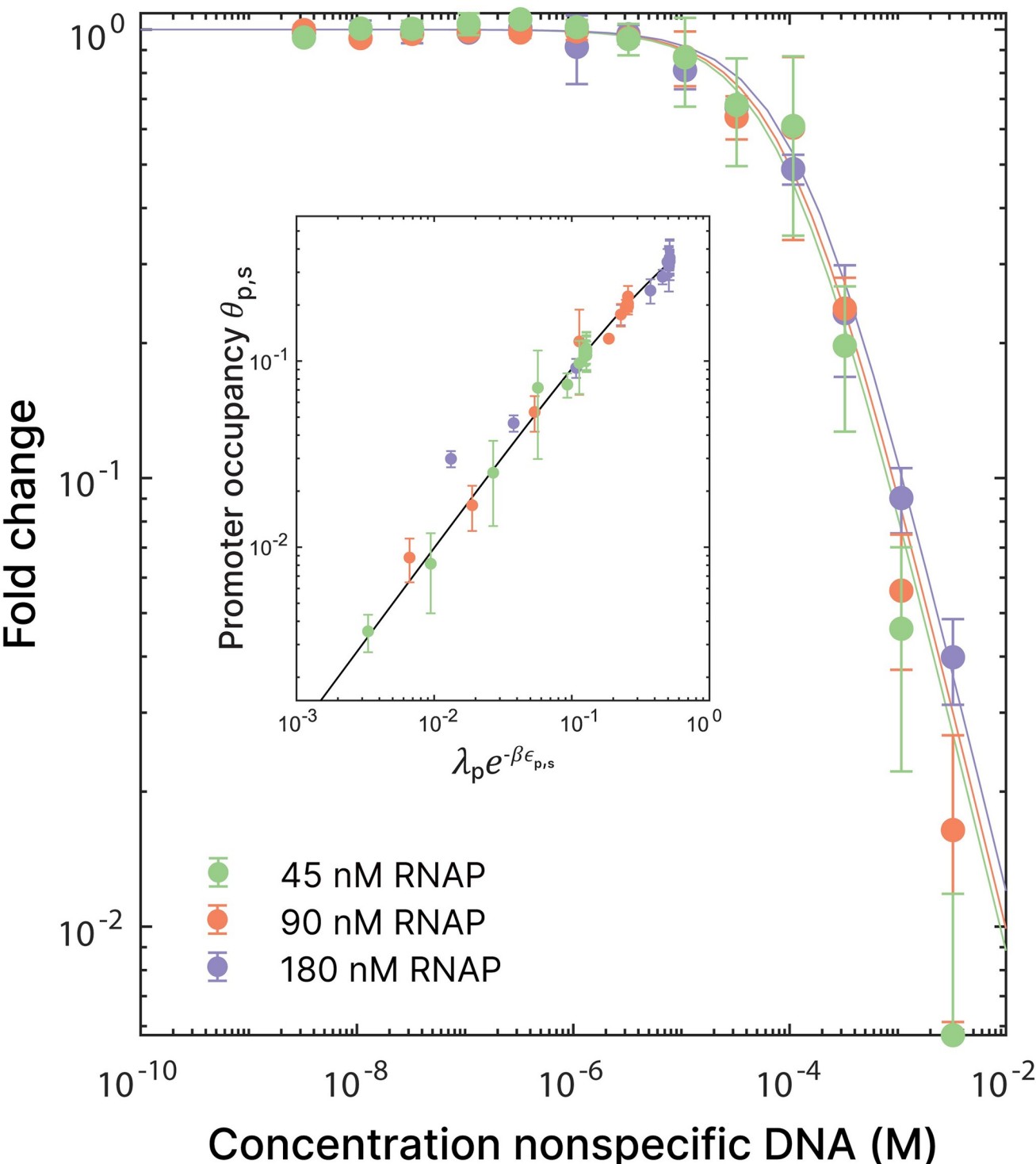

**Fig 4. Nonspecific DNA acts as a repressor.** Master curve for three different RNAP concentrations, with the addition of nonspecific DNA (salmon sperm).

in these studies. Due to cellular processes such as replication and nucleosomal reorganization, the amount of DNA accessible to transcription factors can vary. It is predicted that this effect is nontrivial; even slight changes in transcription factor concentration and variations in the DNA available to bind to might lead to substantial differences in transcriptional outcomes. Hence our choice for these experiments as a case for in vitro transcription studies.

One of the limitations of using the Broccoli aptamer to measure transcription is the folding time of the RNA molecule. It can take more than 10 minutes for the fluorescent signal to transition into the linear regime, as the folding time generates a delay in the signal. This makes it only suitable for thermodynamic measurements over longer times. Regarding the sensitivity of the method, we found that transcriptional activity of 0.5 nM RNAP at a 10 nM DNA concentration was the lowest we could still reliably measure. For reference, a typical transcription-translation reaction using the PURE system contains 100 nM T7 RNAP. The concentration of core RNA polymerases in *E. coli* varies between 2.5 μM and 19 μM depending on the growth conditions [68]. Photobleaching of the fluorescent Broccoli complex has also formed a constraint in studies that use Broccoli as a visualization tool to study transcription dynamics in live cells. In our system, we do not expect fluorophore bleaching to have a major effect, as the fluorescence is only measured in the center of the well. This leaves a significant part of the reaction volume unaffected by irradiation, allowing for efficient replacement of bleached molecules. However, in recent years, improvements have been made to Broccoli-binding fluorophores, enabling higher sensitivity and in vivo cell imaging [16, 69]. Use of these dyes, possibly in combination with Broccoli variants that bind more fluorophores, might improve the method in future studies.

Taken together, we show how the Broccoli aptamer can be used to quantitatively study transcription regulation *in vitro*. By moving away from proteins as reporters for transcription studies, we open up the possibility to study complex genetic circuits more directly. One could think of scenarios with multiple binding sites for various transcription factors, or studies to systematically analyze promoter or protein variants. For novel transcription factors, this method could be used for straightforward quantitative screening. However, it is hard to predict at this moment exactly how far we can push this system. The main results so far show agreement with previous *in vivo* findings, which is a promising starting point for future work. To speculate, improvements in the dye and aptamer might at some point also allow for quantitative *in vivo* studies. This work clearly demonstrates the potential for more complex regulatory scenarios *in vitro*.

## Supporting information

**S1 Text. Derivation of a master curve for *lac* repression.**
(PDF)

**S2 Text. Folding delays of the Broccoli-aptamer.**
(PDF)

**S3 Text. Multi-chemical equilibrium model for tetrameric repression.**
(PDF)

**S4 Text. Template controls.**
(PDF)

**S1 Table. DNA sequences.**
(PDF)

**S1 Data. Data.**
(ZIP)

## Author Contributions

**Conceptualization:** Thomas Visser, Gert Folkers, Willem Kegel.

**Data curation:** Amanda van der Sijs.

**Formal analysis:** Amanda van der Sijs, Pepijn Moerman, Willem Kegel.

**Funding acquisition:** Willem Kegel.

**Investigation:** Amanda van der Sijs.

**Methodology:** Amanda van der Sijs, Thomas Visser, Gert Folkers.

**Supervision:** Gert Folkers, Willem Kegel.

**Validation:** Amanda van der Sijs, Pepijn Moerman, Willem Kegel.

**Visualization:** Amanda van der Sijs.

**Writing – original draft:** Amanda van der Sijs.

**Writing – review & editing:** Gert Folkers, Willem Kegel.

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
