## [Decision Letter · Decision Letter 0]

8 Mar 2024

PONE-D-24-06193Broccoli aptamer allows quantitative transcription regulation studies in vitroPLOS ONE

Dear Dr. van der Sijs,

Thank you for submitting your manuscript to PLOS ONE. After careful consideration, we feel that it has merit but does not fully meet PLOS ONE’s publication criteria as it currently stands. Therefore, we invite you to submit a revised version of the manuscript that addresses the points raised during the review process.

We look forward to receiving your revised manuscript.

Kind regards,

Jianhui Liu

Academic Editor

PLOS ONE

Journal Requirements:

   "Dutch Research Council (NWO) KLEIN grant"

5. In the online submission form, you indicated that "The data can be obtained upon request via email."

7. Please upload a copy of Figures 5 and 6, to which you refer in your text on pages 10-11. If the figure is no longer to be included as part of the submission please remove all reference to it within the text.

Additional Editor Comments:

Reviewer 1 seems a bit too harsh to me (you still need to address), and Reviewer 2-4 comments need your particular attention. 

Reviewers' comments:

Reviewer's Responses to Questions

**Comments to the Author**

1. Is the manuscript technically sound, and do the data support the conclusions?

Reviewer #1: Partly

Reviewer #2: Yes

Reviewer #3: Yes

Reviewer #4: Yes

2. Has the statistical analysis been performed appropriately and rigorously? 

Reviewer #1: Yes

Reviewer #2: Yes

Reviewer #3: Yes

Reviewer #4: Yes

3. Have the authors made all data underlying the findings in their manuscript fully available?

Reviewer #1: Yes

Reviewer #2: Yes

Reviewer #3: Yes

Reviewer #4: No

4. Is the manuscript presented in an intelligible fashion and written in standard English?

Reviewer #1: Yes

Reviewer #2: Yes

Reviewer #3: Yes

Reviewer #4: Yes

5. Review Comments to the Author

Reviewer #1: The manuscript presents detailed description of in vitro transcription depended on the concentration of a repressor and an enzyme. This description is logic and well-written, however, the coincience of the experiment and the model is not sufficient. The model does not predict the RNAP-dependent differences in curves that are clearly seen in Figures 3 and 4. This obstacle indicates some disfunction in the model. I suggest its revision an adding some other step.

Also it is not obvious what is the scientific novelty of the manuscript. Broccoli aptamer is a known tool for transcription visualization in cells. The description of transcription process was also performed a lot of times. It is unclear what new information can we extract from this research.

Reviewer #2: The authors present a quantitative in vitro transcription assay using RNA aptamers coupled to derivations of thermodynamic models of transcription. The approach is quite straightforward and the calculations are rigorous, which is welcome in the field. The results are not revolutionary compared to other available methods, but the authors show that the method follows expected quantitative predictions, with the advantages of fluorescence. They show an interesting example of repression by competitive binding of RNAP.

I have a few remarks regarding clarity and formatting.

1. The introduction contains a whole derivation that should not belong to the introduction (l 47 to 83). It interrupts the presentation of the paper. Consider moving that derivation to the Methods section, or to Results if you consider is as a new theoretical framework that accompanies the data.

2. The calculation is derived in the grand canonical ensemble using "fugacities". The theory seems fine to me. But this framework is hard to follow for anyone who is not an expert in chemical thermodynamics, especially in the gene regulation community. It is the authors' choice to choose their readership. Please at least give the definition of fugacity. Other theoretical models have expressed the expression as a function of RNAP/regulator concentration, using a different ensemble. As a theoretician, I appreciate the rigor exhibited by the authors of this paper; although as a computational biologist focusing on regulation, I do appreciate being able to understand the main equations in terms of experimentally accessible variables such as concentrations, if this does not prevent a rigorous analysis.

3. l 178: "which depends on both the rate of RNA synthesis and the rate of aptamer folding". This sentence does not mention the possible role of fluorophore binding, although it is included in the calculation S2.

4. Delay because of folding. I was surprised by the very slow folding, although the authors mention a comparable reported value. This is a drawback, as mentioned in the Discussion. Maybe the authors should discuss if the in vivo timescales are comparable, and/or if the method seems also applicable in vivo (in terms of fluorescence intensity).

5. l 216: "The key is that...". I don't see what key; the concentrations of RNAP and regulator may be (and in fact are in the cell) hugely different (maybe 1000-fold); so a difference in binding free energy is not conclusive in itself, is it?

6. Fig. 4: The authors mention the concentration values of competitive DNA, but (maybe I missed in the text) the concentration of promoter DNA used. I would assume the competition depends on the ratio promoter/competitor, so this seems important to me but is not mentioned in the Results paragraph nor the figure legend. Or let me know if I am mistaken. Also, I assume that the binding free energy of competitor DNA depends strongly on its sequence, presence of promoter-like sequences, etc? The authors only consider one competitive sequence, so they might warn the reader that the inferred value is specific to this experiment.

Reviewer #3: The paper presents an investigation into transcription regulation in vitro using the fluorescent Broccoli aptamer, focusing on the relationship between RNAP promoter occupancy and transcription rates, the effect of folding delays on transcription rate measurement, the repression mechanism via the LacI repressor, and the impact of nonspecific DNA binding on transcription activity.

While the experimental setup and the data presented are methodologically sound, providing quantitative insights into transcription regulation and the role of nonspecific DNA binding, the paper seems to reiterate findings that align closely with established models and previous studies. The confirmation of proportional relationships between transcription rates and promoter occupancy by RNAP, the quantification of repression mechanisms using LacI, and the influence of nonspecific DNA on transcriptional activity do not deviate significantly from expected outcomes based on existing literature. The data on the influence of nonspecific DNA on transcriptional activity is especially weak.

One of the paper's main contributions is its quantitative approach towards understanding the dynamics of transcription regulation in a controlled in vitro setting, using a novel fluorescence-based method. However, the novel insights for researchers, especially those looking for breakthrough findings or transformative concepts in the field of transcription regulation, may seem limited. The validation of existing models and the reaffirmation of known regulatory mechanisms, while valuable, do not present a significant leap forward in our understanding of transcriptional regulation complexities.

Furthermore, the paper discusses the limitations associated with the use of the Broccoli aptamer, particularly the folding delay impacting the immediacy of transcription rate measurements, which suggests that while the method is innovative, it might not be universally applicable for all types of transcriptional studies, especially those requiring real-time analysis.

In conclusion, the paper provides a solid experimental framework and a detailed quantitative analysis of transcription regulation mechanisms in vitro. It reaffirms established models and offers precise measurements that could serve as a reference for future studies. However, for a research community keen on groundbreaking discoveries and novel insights into the regulation of gene expression, the findings may appear as incremental advancements rather than substantial leaps forward. Future work that builds on these methodologies to explore uncharted territories in gene regulation or introduces innovative experimental designs to capture the dynamic nature of transcriptional regulation might be more impactful for advancing the field.

Main questions:

1. Could the authors elaborate on the novel contributions of their study, especially in the context of how their findings significantly advance our understanding of transcription regulation beyond validating existing models? How does the use of the fluorescent Broccoli aptamer specifically contribute to new insights that were previously unattainable?

2. The authors have provided a quantitative framework for understanding transcription regulation. Can they discuss how their quantitative measurements compare with the predictions of existing models?

3. The manuscript mentions the folding delays associated with the Broccoli aptamer and its impact on transcription rate measurements. How do the authors propose to mitigate these limitations in future studies? Are there alternative approaches or improvements to the Broccoli aptamer method that could minimize these delays?

4. Considering the paper focuses on transcription regulation in an in vitro setting, how do the authors envision the application of their findings in more complex in vivo systems? What are the potential challenges and opportunities in translating these quantitative insights to understand transcription regulation in a cellular context?

5. How do the authors believe their findings could influence the design of future experimental studies in gene regulation? Are there particular aspects of their methodology that could be adapted or improved by other researchers to explore gene expression dynamics more comprehensively?

Others:

Figure resolutions: The main text figures need to have their resolutions increased to match the clarity of the supplementary figures (SFigs).

Line 77 and 79: Replace the incorrect front quotation mark in "adsorbed" with the correct quotation mark.

Line 212: Correct the reference to Fig4B, which does not exist in the manuscript. This should be updated to refer to the correct figure based on the context provided in the manuscript.

Line 217: Change "Fig 4" to "Fig 3" to match the correct figure referenced in the text.

Line 244: Change "Fig 3" to "Fig 4" to correct the figure reference according to the context in the manuscript.

The "o" in the caption for Fig 5 seems to be a typo or an unclear abbreviation. The sentence should likely read, "The fraction of DNA bound to..." without the "o," unless "o" is intended to denote something specific.

Reviewer #4: The recent development of Broccoli and similar RNA aptamers has great potential to visualize RNA biology in real time, and is smaller or more direct than MS2-based systems and those requiring mRNA to be translated to reporter proteins. In this manuscript, the authors have employed Broccoli to quantify numerous aspects of transcription in vitro. These include free energies of specific and non-specific (NS) binding to DNA, the impacts of NS DNA or protein repressors, and kinetics of the reporter itself. Generally speaking, the formulas and logic to their derivation, as well as the experiments themselves, are well-performed. The use of a nearly minimal system is a plus and the 10-30s pulse between fluorescent measurements is indeed above the photobleaching recovery time published by others, though it may be worth commenting if varying the recovery time has any general effects on such readouts.

The Broccoli aptamer folding into a relatively simple secondary RNA structure, or associated binding of DFHBI-IT, having such a long time to reach an excitable conformation is rather intriguing. While it would be helpful to measure the kinetics the compound binding to Broccoli that has had time to fold, work done on the Baby Spinach aptamer and DFHBI would indicate that this indeed is likely to be orders of magnitude more rapid and unlikely to effect the conclusions.

The relatively close free energies of binding to the promoter-driven DNA and non-specific DNA is also an interesting finding.

The PLOS Data Policy may require the plotted values and errors as supplementary tables or available in an online repository.

The temperature of the in vitro transcription should be specified (37°C?), for frame of reference for the kinetics.

Minor text comments:

Figures 5 and 6 are better labelled as supplementary figures.

In line 137, it would be easier to read if the 2 and 3 in Cl2/(HCl)3 were made subscript

The section for "Repression using LacI" quotes Fig. 4 in the text when it should be Fig. 3.

Line 361: capitalize figure

Figure 5 legend: remove the o in "o of DNA"

6. PLOS authors have the option to publish the peer review history of their article (what does this mean?). If published, this will include your full peer review and any attached files.

Reviewer #1: No

Reviewer #2: No

Reviewer #3: No

Reviewer #4: **Yes: **Adam W. Whisnant

---

## [Author Response · Author response to Decision Letter 0]

18 Apr 2024

The responses are included in the "Response to Reviewers" file. Regarding a specific request in the Decision Letter:

The correct grant number (funding information) is OCENW.KLEIN 10085.

---

## [Decision Letter · Decision Letter 1]

16 May 2024

Broccoli aptamer allows quantitative transcription regulation studies in vitro

PONE-D-24-06193R1

Dear Dr. van der Sijs,

We’re pleased to inform you that your manuscript has been judged scientifically suitable for publication and will be formally accepted for publication once it meets all outstanding technical requirements.

Kind regards,

Jianhui Liu

Academic Editor

PLOS ONE

Additional Editor Comments (optional):

Reviewers' comments:

Reviewer's Responses to Questions

**Comments to the Author**

1. If the authors have adequately addressed your comments raised in a previous round of review and you feel that this manuscript is now acceptable for publication, you may indicate that here to bypass the “Comments to the Author” section, enter your conflict of interest statement in the “Confidential to Editor” section, and submit your "Accept" recommendation.

Reviewer #2: All comments have been addressed

Reviewer #3: All comments have been addressed

2. Is the manuscript technically sound, and do the data support the conclusions?

Reviewer #2: Yes

Reviewer #3: Yes

3. Has the statistical analysis been performed appropriately and rigorously? 

Reviewer #2: Yes

Reviewer #3: Yes

4. Have the authors made all data underlying the findings in their manuscript fully available?

Reviewer #2: Yes

Reviewer #3: Yes

5. Is the manuscript presented in an intelligible fashion and written in standard English?

Reviewer #2: Yes

Reviewer #3: Yes

6. Review Comments to the Author

Reviewer #2: My remarks have been addressed, even if in a minimal way. The reviesed manuscript is only slightly modified, but I do not see a reason why not to publish it.

Reviewer #3: (No Response)

7. PLOS authors have the option to publish the peer review history of their article (what does this mean?). If published, this will include your full peer review and any attached files.

Reviewer #2: No

Reviewer #3: No

---

## [Editor Report · Acceptance letter]

21 May 2024

PONE-D-24-06193R1 

PLOS ONE

Dear Dr. van der Sijs, 

I'm pleased to inform you that your manuscript has been deemed suitable for publication in PLOS ONE. Congratulations! Your manuscript is now being handed over to our production team.

Kind regards, 

on behalf of

Dr. Jianhui Liu 

Academic Editor

PLOS ONE